# The *exoS*, *exoT*, *exoU* and *exoY* Virulotypes of the Type 3 Secretion System in Multidrug Resistant *Pseudomonas aeruginosa* as a Death Risk Factor in Pediatric Patients

**DOI:** 10.3390/pathogens13121030

**Published:** 2024-11-22

**Authors:** Carolina G. Nolasco-Romero, Francisco-Javier Prado-Galbarro, Rodolfo Norberto Jimenez-Juarez, Uriel Gomez-Ramirez, Juan Carlos Cancino-Díaz, Beatriz López-Marceliano, Magali Reyes Apodaca, Mónica Anahí Aguayo-Romero, Gerardo E. Rodea, Lilia Pichardo-Villalon, Israel Parra-Ortega, Fortino Solórzano Santos, Mónica Moreno-Galván, Norma Velázquez-Guadarrama

**Affiliations:** 1Laboratorio de Investigación en Microbiología y Resistencia Antimicrobiana, Hospital Infantil de México Federico Gómez, Mexico City 06720, Mexico; carolinagnolascor@gmail.com (C.G.N.-R.); urielgoramirez93@outlook.es (U.G.-R.); bettymar98@gmail.com (B.L.-M.); 2Posgrado en Ciencias Quimicobiológicas, Escuela Nacional de Ciencias Biológicas, Instituto Politécnico Nacional, Mexico City 11350, Mexico; ge_rodm@hotmail.com; 3Departamento de Investigación, Hospital Infantil de México Federico Gómez, Mexico City 06720, Mexico; frjavipg@gmail.com; 4Departamento de Infectología, Hospital Infantil de México Federico Gómez, Mexico City 06720, Mexico; dr.jimenezjuarz@gmail.com (R.N.J.-J.); monica.aguayo@live.com (M.A.A.-R.); 5Laboratorio de inmunomicrobiología, Departamento de Microbiología, Escuela Nacional de Ciencias Biológicas, Instituto Politécnico Nacional, Mexico City 11350, Mexico; jccancinodiaz@hotmail.com; 6Unidad de Investigación y Diagnóstico en Nefrología y Metabolismo Mineral Óseo, Hospital Infantil de México Federico Gómez, Mexico City 06720, Mexico; maghimfg@gmail.com; 7Laboratorio Clínico, Hospital Infantil de México Federico Gómez, Mexico City 067209, Mexico; sudpichis@hotmail.com (L.P.-V.); i_parra29@hotmail.com (I.P.-O.); 8Laboratorio de Investigación en Enfermedades Infecciosas, Hospital Infantil de México Federico Gómez, Mexico City 06720, Mexico; solorzanof056@gmail.com; 9Departamento de Hemato-Oncología, Hospital Infantil de México Federico Gómez, Mexico City 06720, Mexico; monikmg1970@yahoo.com.mx

**Keywords:** type 3 secretion system, death, multidrug resistance, *Pseudomonas aeruginosa*

## Abstract

The poor prognosis of infections associated with multidrug-resistant *Pseudomonas aeruginosa* can be attributed to several conditions of the patient and virulence factors of the pathogen, such as the type III secretion system (T3SS), which presents the ability to inject four effectors into the host cell: ExoS, ExoT, ExoU and ExoY. The aim of this study was to analyze the distribution of *exo* genes through multiplex polymerase chain reaction in *P. aeruginosa* strains isolated from patients at a third-level pediatric hospital and their relationships with clinical variables, e.g., the origin of the sample, susceptibility profile and outcome, through a multinomial logistic regression model. A total of 336 bacterial strains were obtained from cystic fibrosis (CF; n = 55) and bloodstream infection (BSI; n = 281) samples, and eleven presence (+)/absence (−) *exo* virulotype patterns were identified. The virulotype V3 (*exoU−/exoS+/exoT+/exoY+*) was observed in 64.28%, followed by V1 (*exoU+/exoS−/exoT+/exoY+*) with 11.60%. Additionally, V2 (*exoU+/exoS−/exoT+/exoY−*) was present in 11.60%, and V7 (*exoU−/exoS+/exoT+/exoY−*) was present in 4.17%. The remaining virulotypes (8.33%) identified were clustered in the other virulotype (OV) group (V4, V5, V6, V8, V9, V10 and V11). The clinical records of 100 patients and their outcomes were reviewed. Fifteen patients died (CF = 4; BSI = 11). V2 and V1 were the virulotypes most related to pandrug resistance (PDR), whereas the V1 relative risk of death was determined to be almost four-fold greater than that of V3, followed by V2 and OV. In summary, the virulotypes V1, V2 and CF are related to death. This study highlights the association of T3SS virulotypes with the susceptibility profile, clinical origin and their potential for predicting a poor prognosis.

## 1. Introduction

Infections caused by multidrug-resistant and extensively drug-resistant (MDR and XDR) *Pseudomonas aeruginosa* are associated with high morbidity and mortality rates, elevated costs and chronic infections, and a poor prognosis is commonly associated with the health status of the patient (i.e., immunocompromised states) and the characteristics of the microorganism, e.g., its resistance phenotype and virulence factors [1,2]. In particular, outcomes from respiratory diseases and *P. aeruginosa* bloodstream infection (BSI) have been attributed to the toxins secreted by the type III secretion system (T3SS) [3]. Four main effectors, which are injected into host cells through the T3SS, have been described in *P. aeruginosa:* ExoS, ExoT, ExoU and ExoY [4]. The most widely characterized and studied among them are ExoS and ExoU, owing to their important clinical relevance [5].

Compared to other common hospital pathogens such as *Staphylococcus aureus* and *Klebsiella pneumoniae*, *P. aeruginosa* has the ability to evade the immune system and cause tissue damage through T3SS effectors, which may represent an additional risk factor in healthcare-associated settings and in areas of immunocompromised patients in adults and children [3,6].

The successful dissemination of MDR/XDR strains of *P. aeruginosa* is influenced by a complex interplay of pathogenicity, epidemic potential and antibiotic resistance, and most of these strains belong to a limited number of globally prevalent clones [7]. A recent study on T3SS in *P. aeruginosa* strains with cytotoxic phenotypes revealed its association with the *exoU* genotype, related to greater capacity for acute tissue damage and sepsis, while non-cytotoxic strains are more frequently related to the *exoS* genotype [8]. An example of this is the Liverpool epidemic disease (LES) *P. aeruginosa* strain ST146, which is an MDR clone that chronically infects and colonizes the lungs of patients with cystic fibrosis [7]. While the *exoU+/exoS−* genotype is an independent risk factor for early mortality in bloodstream infections and shows a negative correlation with XDR profiles, it is also related to the high-risk clone ST235, which is often associated with a poor prognosis [9].

The *exoU* gene is located in different conjugative elements, such as the PAPI-2 pathogenicity island or the ExoU, A, B and C islands, which are integrated through recombination with tRNA-Lys. This gene encodes a potent A2-family phospholipase, which is considered the most virulent effect residing in the T3SS. Its overregulation is correlated with an unfavorable prognosis and early mortality [5,10].

The ExoS protein presents distinct functions, including its GTPase activity over small GTPases, which regulate diverse cellular functions, such as cell migration, phagocytosis and cytoskeleton organization, such as Rho, Rac and Cdc42, and its adenosine diphosphate ribosyl transferase (ADPRT) activity over several Ras-related proteins involved in the organization of the cytoskeleton, which is dependent on the eukaryotic cofactor factor activating ExoS (FAS) [11]. Both activities affect the organization of actin in the cytoskeleton of the host cell by reducing cell-cell adherence and facilitating the invasion of *P. aeruginosa* through epithelial barriers [12]. The *exoS* gene commonly presents distributions excluding the *exoU* gene, whereas the *exoT* and *exoY* genes are present in most isolates and are considered part of the core genome of the bacterium [5,13].

The mutual exclusion of ExoS and ExoU is difficult to interpret because of the chromosomal codification of *exoS* and the presence of *exoU* in a mobilizable genetic element [5,13,14]. Additionally, the expression of *exoU* and *exoS* in isolates that present or lack both genes is uncommon. Although both exotoxins facilitate several mechanisms for bacterial propagation and pathogenesis, these effectors are not expressed cooperatively or simultaneously; however, while the secretion of ExoS has been associated with endocytosis and intracellular bacterial survival, ExoU induces rapid destruction of the plasmatic membrane of host cells [6,12,15].

On the other hand, recent studies have shown that ExoT induces cell cycle arrest in the G1 phase in melanoma cells, suggesting that this effector plays a regulatory role in the cell cycle [16]. ExoY presents nucleotide cyclase activity. This effector promoted the accumulation of different cyclic nucleotides (cNMPs), with a preference for guanine monophosphate (cGMP) and uracil monophosphate (cUMP) over adenine monophosphate (cAMP) and cytokine monophosphate (cCMP); however, its contribution to the virulence of *P. aeruginosa* is still unclear [17].

*P. aeruginosa* strains isolated from early infections have shown a higher frequency of T3SS-secreted effectors, followed by isolates from children and, finally, adults with chronic infections. Furthermore, an inverse correlation is observed between the duration of infection and the decrease in the percentage of isolates secreting T3SS effectors [18].

Several methods for the molecular detection and identification of the *P. aeruginosa* T3SS *exoU*, *exoS*, *exoT* and *exoY* virulotypes have been widely implemented through multiplex polymerase chain reaction (mPCR) [19]; nevertheless, the relationship between the complete genotype and its impact on the prognosis of infection has not been completely established. In the case of infections caused by MDR *P. aeruginosa*, a poor prognosis can be attributed to several medical conditions of the patient, highlighting clinical cases with preexisting comorbidities, in addition to the presence of virulence factors, such as the T3SS, which plays a crucial role in the pathogenicity of the microorganism [10]. In the present study, we analyzed the presence of the genes coding for exotoxins and their multiple combinations in isolates of multidrug-resistant *P. aeruginosa* recovered from pediatric patients at a third-level pediatric hospital and their relationships with specific variables, such as the origin of the clinical sample used for pathogen isolation, the susceptibility profile of the isolates and the clinical outcome of the patients.

## 2. Materials and Methods

### 2.1. Collection of Biological Samples

A total of 336 cultures of *P. aeruginosa* from cystic fibrosis (CF; n = 55) and blood-stream infection (BSI; n = 281) from pediatric patients at the Hospital Infantil de México Federico Gómez (HIMFG) were obtained from 2017 to 2023. The samples (blood and sputum) from the Department of Central Laboratory of the HIMFG were kept at 4 °C for no more than three days for immediate use. The bacterial cultures were collected as follows: Clinical isolates were initially inoculated in Petri dishes with blood agar (DIBICO). The plates were incubated at 37 °C for up to 96 h. After bacterial growth, reseeding was performed on both Mueller-Hinton (BIOXON) and cetrimide (OXOID) differential and selective agar plates, which were subsequently incubated for 48 h at 37 °C and 42 °C, respectively, for identification according to colony and microscopic morphology. The production of N2 in nitrate broth (DIBICO) supplemented with Durham tubes, oxidase and catalase activities, pigment production and growth at 42 °C were also evaluated. Bacterial identity was confirmed with a MALDI-TOF (Biomerieux, Marcy l’Etoile, France) automated system. All strains were preserved in silk milk (OXOID) at −70 °C for further use. This study was approved by the HIMFG Research, Biosafety and Ethics Committees (HIM/2017/042. SSA 1742). In the latter, the need for informed consent and/or assent was assessed based on the age of the patient. Informed consent was signed by the patient’s parents or representatives; only *P. aeruginosa* isolates from bloodstream infections (BSI) or cystic fibrosis (CF) were requested. Patient identification data were collected solely by the committee and researchers under its supervision. The only data required for the study were the source of the sample and the outcome.

### 2.2. Molecular Detection of Exo Virulotypes Through mPCR

The isolation and purification of genomic DNA from the cultures confirmed as the species *P. aeruginosa* were performed as follows: tubes with Mueller-Hinton broth were inoculated with the bacterial isolates. All the tubes were incubated at 37 °C for 18 h. The biomass was processed with the Wizard^®^ Genomic DNA Purification Kit (Promega, WI, USA.) according to the manufacturer’s instructions.

The design was performed on the sequences of the *exoS* (NP_252530.1), *exoT* (NP_248734.1), *exoY* (NP_250881.1) and gyrB (NP_064724.1) genes of strain PAO1 (NC_002516.2) and the *exoU* gene (NC_008463.1) of strain PA14 deposited in GenBank. The primers used were designed in silico using the Primer-BLAST server (NCBI) and the gene set was selected according to homology, GC% content, melting temperature (Tm) and amplicon size generated [19,20,21,22,23] (Table 1).

The reactions were aliquoted in a final volume of 50 µL, containing 25 µL of 2X Multiplex PCR Master (JenaBioscience^®^, Jena, Germany), 10 µM forward and reverse primers and 50 ng of genomic DNA, according to the manufacturer’s instructions. The enzymatic reactions were performed in an Axygen thermal cycler (Corning, MA, USA) under the following conditions: initial denaturation at 95 °C for 2 min; 40 cycles of denaturation at 95 °C for 15 s, annealing at 62 °C for 1 min and extension at 72 °C for 40 s; and a final extension cycle at 72 °C for 5 min, according to the manufacturer’s instructions, and the annealing temperatures of each pair of primers designed in this study were considered.

The presence (+)/absence (−) of *exo* genes in the bacterial isolates was assessed in order to analyze the distribution of *exo* genes in the 336 isolates. The bacterial strains *P. aeruginosa* PAO1 and PA14 were employed as positive controls for the *exoS* and *exoU* genes, respectively. Both strains amplify *exoT* and *exoY*, in addition to the constitutive gene *gyrB* of *P. aeruginosa*. Blood and CF samples positive for *Staphylococcus aureus*, *Stenotrophomonas maltophilia*, *Acinetobacter baumannii* and *Klebsiella pneumoniae* were used as negative controls (Appendix A)

The PCR products were finally loaded onto a 1% agarose gel supplemented with Midori Green Advance (NipponGenetics, Tokyo, Japan). The running conditions for electrophoresis were determined as follows: 90 V, 300 mA, per 40 min. Amplicons were visualized in a Syngene GVM20 UV transilluminator.

### 2.3. Determination of the Susceptibility Profile

The susceptibility profiles of 105 isolates were previously reported by Aguilar-Rodea et al. [24,25]. The remaining isolates were evaluated to determine their susceptibility profile by employing the minimum inhibitory concentration (MIC) method via the agar dilution test, according to the Clinical Laboratory Standard Institute (CLSI) 2024 edition guidelines and breakouts established for *P. aeruginosa* [26]. Thirteen antibiotics (Sigma-Aldrich, St. Louis, MO, USA) were evaluated: gentamycin (GEN), tobramycin (TOB), amikacin (AK), meropenem (MEM), imipenem (IMI), ceftazidime (CAZ), cefepime (CPM), ciprofloxacin (CIP), levofloxacin (LEV), piperacillin/tazobactam (P/T), aztreonam (AZT), fosfomycin (FOS) and colistin (CS). The reference strains employed for validation of the test included the *Pseudomonas aeruginosa* ATCC 27853 and the *Escherichia coli* ATCC 25922 (American Type Culture Collection, Manassas, VA, USA) reference strains. Since the 2024 edition of the CLSI guidelines does not include breakout points for the evaluation of the susceptibility of fosfomycin (FOS), the values proposed by Smith et al. [27] were employed, which were followed up by Aguilar-Rodea et al. [25]. The cutoff points are as follows: S ≤ 64 µg/mL, I = 128 µg/mL and R ≥ 256 µg/mL. Additionally, the 2024 CLSI edition eliminates the breakpoints for gentamicin, so the breakpoints to be used for this antibiotic correspond to those present in the 2023 edition, which are S ≤ 4 µg/mL, I = 8 µg/mL and R ≥16 µg/mL [28]. Categorization of resistance according to Magiorakos et al. [29] was employed for classification of the isolates studied as multidrug resistant (MDR; nonsusceptible to at least one agent in three or more categories of antibiotics), extensively drug resistant (XDR; nonsusceptible to at least one antimicrobial agent in all but one or two categories), or pandrug resistant (PDR; resistant to all categories of antibiotics) [29].

### 2.4. T3SS Virulotypes and Clinical Variables

For this study, we considered samples from pediatric patients with a confirmed CF diagnosis and positive *P. aeruginosa* culture, in addition to blood culture samples positive for the bacteria. Initial analyses sought to evaluate the T3SS virulotypes as the response variable, while predictor variables included sample origin (BSI/CF), susceptibility profile and patient outcome. Ultimately, 137 isolates of the initial 336 that met these criteria were included.

To obtain the relationship between virulotypes and clinical outcome, the response variable was patient mortality, operationally defined as death attributed to bacterial infection in cases of cystic fibrosis (exacerbation due to *P. aeruginosa* infection) and, in cases of bloodstream infection, bacteremia (isolation of *P. aeruginosa* in cases of primary bacteremia on days 7, 14 and 30) and sepsis (presence of tachycardia, pyrexia and hypotension due to infection). Duplicate data from the same patient were discarded and only the isolate associated with death was considered, resulting in 100 cases. In this data, clinical death was attributed to bacterial infection in four CF patients and eleven BSI cases.

### 2.5. Statiscal Analysis

For the first analysis, data were summarized overall and by virulotypes. Categorical variables were described as frequency and percentages, and Fisher’s exact test was used. A multinomial logistic regression model was developed to associate virulotypes within the clinical data of the analyzed samples with “V3” as the reference class, and each covariate was quantified by generalized odds ratios (gORs). The inclusion criteria were as follows: bacterial isolates of clinical origin (i.e., CF, BSI); evaluation of isolates to determine their susceptibility profile, categorized according to Magiorakos et al. [29]; and clinical records documenting either discharge or death attributed to bacterial infection.

In a subsequent set of analyses, mortality was analyzed as a response variable. A chi-square test or Fisher’s exact test was performed to assess the presence of an association between 2 categorical variables. Factors associated with mortality were identified using a multilevel Poisson regression model with robust error variance (level 1: strains and level 2: patients).

Statistical analysis was performed with Stata version 17.0 (Stata, Stata Corp, College Station, TX, USA) and RStudio version 4.4.0 (Vienna, Austria) packages [30,31]. A *p*-value of <0.05 was considered statistically significant.

## 3. Results

### 3.1. Identification of T3SS Virulotypes

The previously designed primers were used to standardize the reaction. The annealing temperatures of each pair of primers were taken into account with the objective of assessing the correct identification of each gene, avoiding the formation of secondary products and false positives and optimizing the reaction without losing its effectiveness. Once the optimal amplification conditions were established, the exo genes were screened in the collection of clinical isolates.

The distribution of the *exo* genes in the *P. aeruginosa* isolates in this study revealed that the *exoU* gene was present in 24.70% (n = 83/336) of the isolates, whereas the *exoS* gene was present in 69.94% (n = 235/336) of the isolates and *exoT* was identified in 96.13% (n = 323/336) of the isolates. Additionally, *exoY* was observed in 80.65% of the evaluated isolates (n = 271/336). Up to eleven virulotypes patterns corresponding to the T3SS *exo* genes were identified in the *P. aeruginosa* isolates. The most abundant pattern was virulotype V3, which was observed in 63.98% of the isolates (n = 215/336), followed by virulotype V1, which was identified in 11.60% (n = 39/336); virulotype V2, which was identified in 11.60% (n = 39/336) of the isolates; and V7, which was identified in 4.16% (n = 14/336) of the isolates. For analysis, the remaining virulotypes identified in the study were identified and clustered into the other virulotype (OV) group (V4, V5, V6, V8, V9, V10, V11), which accounted for up to 8.63% (n = 29/336) of the total (Table 2; Appendix A).

### 3.2. Associations Between Multidrug Resistance and Virulotypes

We analyzed the association between the different virulotypes and origin of the sample, susceptibility to antibiotics and patient outcome. Those data that did not have all this information was discarded, leaving 137 that met these criteria. A total of 113 (82.48%) isolates were obtained from BSI patients. The remaining isolates were obtained from CF patients (16.06%; n = 22). According to the categorization of resistance of isolates, 20.44% (n = 28/137) of the cultures were categorized as PDR, 46.72% (n = 64/137) were classified as XDR and 28.49% (n = 39/137) were determined as MDR (Table 3, Appendix A). Regarding the clinical outcome and its consideration of all the isolates obtained from one single patient, the deaths of 15 patients were attributed to bacterial infection; 4 of these cases were diagnosed with CF (V3 in 6 cases) and 11 were diagnosed with BSI (dominance of V1 and V2 in 5 and 3 cases, respectively).

In our dataset, the most frequent T3SS virulotypes were V3, with a frequency of 53.28% (n = 73/137); V2, with 24.08% (n = 33/137); V1, with 11.67% of the isolates (n = 16/137); and V7, with 2.18% (n = 3/137). The remaining virulotypes (OVs) were identified with a frequency of 8.75% (n = 12/137). The model employed the most frequent virulotype as a reference, which was represented by V3. Baseline characteristics of the sample are shown stratified by virulotypes (Table 3). The highest rate of mortality was observed for V1 (37.5%). Regarding clinical origin, V2 was detected with a significantly higher frequency among isolates from BSI, as was V7 from CF (*p* = 0.001). Finally, OV, V1 and V3 were more predominant with an XDR profile, as was V2 with a PDR profile (*p* < 0.001).

In the multinomial logistic regression analysis (Table 4), isolates from patients with death showed higher odds of having V1 (gOR = 5.120, 95% CI: 1.264–20.733) and V2 (gOR = 4.756, 95% CI: 1.113–20.328) versus having V3. Additionally, CF patients had lower odds of having V1 versus having V3 compared with BSI patients (gOR = 0.16, 95% CI: 0.025–0.999). Finally, isolates with a PDR profile presented greater odds of having V2 (gOR = 38.790, 95% CI: 8.319–180.895) versus having V3 compared with isolates with MDR, followed by the OV (gOR = 14.890, 95% CI: 1.079–205.387) and V1 (gOR = 7.000, 95% CI: 1.040–47.099) groups. Isolates with XDR had a higher odds of having OV versus isolates with MDR (gOR = 10.506, 95% CI: 1.212–91.053).

### 3.3. T3SS Virulotype as a Risk Factor for Death

The relationships between the study variables (clinical origin of the isolate, susceptibility profile and T3SS virulotype) and death are described in Table 5, with the V1 virulotype and the XDR resistance phenotype having significant values.

In the model where the response variable was death, a total of 15 cases were used to determine the outcome of death caused by the infection. A total of 4 patients were diagnosed with CF, while the remaining 11 patients were diagnosed with BSI.

Table 6 presents the factors associated with mortality. Compared with the V3 virulotype, the V1 virulotype had a 3.69-fold increased risk of death (RR = 3.690, 95% CI: 1.259–10.82). Compared with V3, V7 (RR = 0.00000001, 95% CI: 0–0.0000001) presented a negative association with death. Patients with CF had a greater risk of death than those with BSI (RR = 3.3643, 95% CI: 1.181–11.230).

## 4. Discussion

The T3SS is a virulence factor consisting of a needle-like multicomponent complex that injects effector proteins from the cytosol of bacteria into the host cell and has been associated with poor prognosis in cases of acquired pneumonia, keratitis and otitis [15]. In the case of *P. aeruginosa*, the T3SS is associated with the severity of infection and has been demonstrated when it allows the injection of Exo effectors [3,4,5].

By developing a mPCR assay, it was possible to analyze the frequency of the virulotypes evaluated to reduce costs and make the determination quickly [21,22,23]. The distribution of each *exo* gene in the 336 *P. aeruginosa* isolates recovered from pediatric patients in the HIMFG were evaluated, and the results indicate that the *exoT* gene was the most common (96.13%), whereas *exoU* was considered the least common. In this work, more CF isolates were positive for the *exoT* and *exoS* genes, and only two positive cases were positive for *exoU.* Similar data were reported by other groups in CF, who reported the *exoT* gene in 47 isolates (95.9%) (n = 47/49), whereas both *exoY* and *exoS* were identified in 48 isolates (97.9%) (n = 48/49); however, *exoU* was identified in 31 CF isolates (63.2%) (n = 31/49) [32]. Other studies reported that the *exoS* gene was the most prevalent gene, followed by *exoU*, with lower frequencies of both *exoT* and *exoY* [33].

On the other hand, it has been suggested that the presence of the *exoU* gene varies significantly according to the origin of the sample, e.g., its presence ratios are observed in bacteremia (13%–64%), CF (10%) and keratitis (61.5%) [34]. Otherwise, the study performed by Juan et al. [3] reported the *exoS* gene at frequencies ranging from 58 to 72% [3]. They associated this effector with an invasive phenotype, as observed with PAO1 or PAK strains. Additionally, they reported lower frequencies of *exoU* (28–42%) and its association with highly cytotoxic phenotypes, as observed with PA14 or PA103 strains [3]. According to our results, the sole presence of *exoU* is not associated with a specific origin of isolation (BSI; CF). A similar trend was demonstrated for the prevalence of genes encoding effector proteins of the T3SS, which was independent of the site of infection, with the exception of the isolate recovered from the CF samples, where *exoS* was prevalent [35].

The calculated models demonstrated that the relative risk of death was greater in virulotype V1, where *exoS* was absent. Moreover, the *exoU+* gene has been associated with death in early bacteremia; recent studies also associated this gene with multidrug resistance events, highlighting resistance to fluoroquinolones [3,9,14,36,37]. Furthermore, it was possible to identify the relative risk for death regarding the T3SS virulotypes V1 and V2, which presented an *exoU+* genotype. Additionally, V2 presented a relatively high risk of presenting a PDR resistance phenotype, followed by V1, whereas V3 presented a relatively high probability of presenting an MDR resistance phenotype.

Some studies have explored the phenotypic secretion of T3SS effectors simultaneously in *P. aeruginosa.* Through immunoassays, the prevalence of ExoU/ExoT-secreting isolates was identified compared with the ExoS/ExoT phenotype in respiratory tract clinical samples, as was the expression of ExoT and ExoS alone with no other exotoxins. In addition, none of the isolates expressed ExoU alone [38]. Although expression assays were not performed in this study, virulotypes with the sole presence of *exoU−* or *exoU/exoY* positive variants without the presence of other exotoxins were not identified. Instead, the V7 virulotype was present in only 3.91% (n = 14/336) and did not present a relative risk for death according to the results of the models.

Some limitations of our study are the lack of identification of chaperone proteins and the evaluation of the effects of these exotoxins in both in vitro and in vivo models. Wu et al. [34] demonstrated, through in silico analysis, the association of the loss of cytotoxicity with the presence of a punctual mutation in the SpcU chaperone protein, which is critical for the optimal activity of ExoU [34]. Additionally, studies on the structure of T3SS have shown that alterations in its architecture result in a decrease in virulence, despite the presence of toxic effectors [5]. However, the data generated in this work reveals a virulotype-prognosis relationship that can be investigated considering these factors in further studies. Other aspects to consider in the present study are the fact that samples were obtained from a single pediatric center, which indicates that our results reflect the behavior of T3SS virulotypes in a delimited sector of the pediatric population, of which, in addition, other clinical conditions could be considered for future studies that allow establishing the relationship of these patterns with the outcome and conditions of patients (i.e., hematological diseases that increase the risk of death [39]). In addition, the relationship of other virulence factors that contribute to the pathogenicity of the microorganism should be considered.

One of the main differences among the virulotypes identified in our study was the presence of the exoY gene in V1, V3, V5, V8 and V10, with distinct frequencies ranging from 0.29% to 63.98%. No differences in prevalence were observed regarding the presence or absence of the exoY gene in V1 and V2, respectively; however, both virulotypes presented 5.1- and 4.7-fold increased risks of death with respect to V3, where *exoU* was absent and *exoY* was present. In contrast to the results reported by other researchers who identified *exoU/exoY* genotype clinical isolates, a close relationship among the *exoY+* genotypes, in addition to lung damage in cases of nosocomial pneumonia, was demonstrated [40]. Other research groups have also demonstrated that the *exoY* gene presents a high rate of nonsynonymous single-nucleotide polymorphisms (SNPs) and deletions, especially in clinical isolates. However, the changes produced by SNPs and deletions (frame shifts and protein truncation) were not sufficient to prevent infection, considering that other virulence factors could be coexpressed at the time of infection. [8,19].

Several studies have analyzed the T3SS virulotype, focusing on the presence of ExoS and ExoU, which are considered the most prevalent effectors and have been employed as typing genes for *P. aeruginosa* high-epidemiological-risk clones [8,36,37,38]. In the present study, the 105 *P. aeruginosa* strains previously typed by multilocus sequence typing (MLST) and the genotyping data of the MexAB-OprM pump regulators, described by Aguilar-Rodea et al. [24,25] (2017), were considered. These data are part of the working group collection (Appendix A). No significant relationship was identified between T3SS virulotyping and sequence type (ST235, ST233, ST1725), haplotype of MexAB-OprM (H12), including the origin of the sample, susceptibility profile and clinical outcome. These clones are related to XDR/MDR resistance phenotypes and *exoU+* and *exoS+* genes, forcing us to investigate the potential of these isolates to clonally disseminate [24,41].

The high-epidemiological-risk clone ST235 presented a significant rate of the presence of the exoU gene [14]. Its combination is unfavorable for prognosis. Moreover, this clone has been demonstrated to present increased mortality in comparison with other high-risk clones [14,36]. Clones related to CC235 presented both V1 and V2 virulotypes in four clinical cases, with a clinical outcome of death. Even though the WHO reclassified *P. aeruginosa* as a high-priority group [42], higher-death-risk clones are distributed worldwide, representing a health issue in patients with infections associated with these strains. This evidence highlights the need for characterization of every clinical isolate, from evaluation of the susceptibility profile, genotyping of the T3SS and, if possible, characterization of the MexAB-OprM efflux pump haplotype. The application of these measures in the early clinical diagnosis of infection must still be considered for a favorable and timely response against the bacterial pathogen, especially in high-epidemiological-risk clones and immunocompromised patients.

ExoU-producing *P. aeruginosa* strains have been identified in five-year-old pediatric patients with CF infections; however, these types of strains are poorly represented in persistent chronic infections [13,32,43]. In this work, the isolates recovered from the CF clinical samples presented a dominance of virulotype V3; nonetheless, ten isolates from this group presented *exoU*, which is included in the V1, V2 and V9 virulotypes, possibly attributed to the characteristics of the treatment with antibiotics. Specifically, long-term therapies can induce a selection event in strains, eradicating those presenting *exoU*, whose deficiency can lead to evasion of the immune response, promoting a chronic infection process [13]. In addition, the V9 virulotype, characterized by the presence of the *exoU+/exoS+/exoT+/exoY+* genes, was identified in two isolates from patients with cystic fibrosis (CF). In a study by Sarges et al. [32], the *exoS+/exoU+* virulotype was detected in 61.2% (n = 30) of the isolates evaluated. The authors observed a high frequency of this virulotype in cases of intermittent infection, mainly in the early phases of infection. In contrast, cases of chronic infection and greater severity were associated with the *exoS+* genotype [32]. In a study conducted by Horna et al. [44], an unusually high frequency of the exoU+/exoS+ genotype was recorded. This was attributed to the presence of specific pressures that allow the presence of both genes. Additionally, an association was reported between this genotype and MDR/XDR profiles, which exhibit a tendency toward high levels of resistance to fluoroquinolones [44].

Furthermore, as ExoS inhibits the synthesis of interleukins by alveolar macrophages via the modulation of the initial inflammatory response, a selective event of *exoS+* strains with the capacity to colonize and persist in chronic infectious processes caused by this bacterium can occur [2,4]. Bacterial persistence in chronic infections caused by *P. aeruginosa* can also induce the dysregulation of toxic effectors and the continuous production of antibodies against the T3SS, indicating a role over time in the pathogenesis of the infected lung; for example, in coinfection cases in patients with CF, commonly with *S. aureus*, it has been reported that *P. aeruginosa* does not exclusively employ its T3SS to modulate the immune response of the host and avoid its own eradication but also to eradicate *S. aureus* [4,14,32]. In patients with hematologic conditions, *P. aeruginosa* BSI has been linked to a significantly elevated risk of mortality compared to other Gram-negative bacteria. Similarly, a correlation has been observed between prognosis and the primary site of infection, with mortality rates typically high when associated with pulmonary infection [45]. Conversely, in non-CF isolates, *P. aeruginosa* has been associated with an increased risk of mortality, prolonged hospitalizations and exacerbations in adults with bronchiectasis [46].

In the Mexican pediatric CF patient population, a high prevalence of *P. aeruginosa* colonization has been reported, with 20% of MDR clones and 8.2% of XDR identified [47]. On the other hand, *P. aeruginosa* bacteremia (BSI) is mostly hospital-acquired, with a significant proportion in intensive care units (ICUs) [45]. Mortality associated with *P. aeruginosa* BSI in children is high, with in-hospital mortality rates ranging from 29% to 39% [39,45]. Factors associated with increased mortality include neutropenia, ICU admission and inadequate empiric therapy [45]. In patients with MDR *P. aeruginosa*, an overall mortality rate of 31% has been observed, with higher rates observed in those infected [48]. Numerous studies have shown that appropriate therapy can significantly reduce mortality [9,39], so clinical application of T3SS is relevant.

## 5. Conclusions

According to the results of our study, we can conclude that the *P. aeruginosa* V1 and V2 virulotypes present a greater risk of death. Compared with the V3 virulotype, V2 is mostly involved in bloodstream infections. This study highlights the effects of different combinations of T3SS virulotypes on bacterial virulence, their associations with the susceptibility profile and their potential for predicting a poor prognosis.

## Figures and Tables

**Table 1 pathogens-13-01030-t001:** Primers employed for endotoxin detection via mPCR.

Target Genes	Sequences (5′ to 3′)	Product Size (bp)
*exoU*	F: GAC AGA TCG CTA CGC ATC CA R: AGA TGT TCA CCG ACT CGC TC	688
*exoS*	F: CGT CGT GTT CAA GCA GAT GG R: GAA TGC CGG TGT AGA GAC CA	533
*exoT*	F: TGC GGT AAT GGA CAA GGT CG R: AAC AGG GTG GTT ATC GTG CC	459
*exoY*	F: GTC TCT ACA GGA TCA GCC GC R: CGT CGC TGT GGT GAA ACA TC	330
*gyrB*	F: TGG GAA CAG GTC TAC CAC CA R: CAG ACC GCC TTC GTA CTT GA	243

**Table 2 pathogens-13-01030-t002:** T3SS virulotypes in *P. aeruginosa* clinical isolates.

Virulotype	T3SS Virulotypes	Frequency (n = 336/100%)
V1	** *exoU* ** *+/exoS−/**exoT+**/**exoY**+*	39 (11.60%)
V2	** *exoU+* ** */exoS−/**exoT+**/exoY−*	39 (11.60%)
V3	*exoU−/**exoS+/exoT+/exoY+***	215 (63.9%)
V4	*exoU−/exoS−/**exoT+**/exoY−*	3 (0.89%)
V5	*exoU−/exoS−/**exoT+/exoY+***	8 (2.38%)
V6	*exoU−/exoS−/exoT−/exoY−*	9 (2.67%)
V7	*exoU−/**exoS+/exoT+**/exoY−*	14 (4.16%)
V8	*exoU−/exoS−/exoT−/**exoY+***	3 (0.89%)
V9	** *exoU+/exoS+/exoT+/exoY+* **	4 (1.19%)
V10	*exoU−/**exoS+**/exoT−/exo**Y**+*	1 (0.29%)
V11	** *exoU+/exoS+/exoT+/* ** *exoY−*	1 (0.29%)

Genes that were identified (+) in the virulotypes are highlighted in **bold**.

**Table 3 pathogens-13-01030-t003:** Clinical variables evaluated for the study *P. aeruginosa* isolates.

		Virulotypes						
Variable		OV	V1	V2	V3	V7	Total	*p*-Value
Death	No	10 (90.91%)	10 (62.5%)	27 (81.82%)	63 (85.14%)	3 (100%)	113 (82.48%)	0.274
	Yes	1 (9.09%)	6 (37.5%)	6 (18.18%)	11 (14.86%)	0 (0%)	24 (17.52%)	
Clinical origin	BSI	11 (100%)	14 (87.5%)	33 (100%)	55 (74.32%)	2 (66.67%)	115 (83.94%)	0.001
	CF	0 (0%)	2 (12.5%)	0 (0%)	19 (25.68%)	1 (33.33%)	22 (16.06%)	
Susceptibility profile	MDR	1 (9.09%)	3 (18.75%)	4 (12.12%)	30 (40.54%)	1 (33.33%)	39 (28.47%)	<0.001
	PDR	2 (18.18%)	3 (18.75%)	18 (54.55%)	5 (6.76%)	0 (0%)	28 (20.44%)	
	S	0 (0%)	1 (6.25%)	0 (0%)	4 (5.41%)	1 (33.33%)	6 (4.38%)	
	XDR	8 (72.73%)	9 (56.25%)	11 (33.33%)	35 (47.3%)	1 (33.33%)	64 (46.72%)	

*p*-value < 0.05, BSI: bloodstream infection, CF: cystic fibrosis, PDR: pandrug resistant, XDR: extensively drug resistant, S: sensitive. V1 (*exoU+/exoS−/exoT+/exoY*+); V2 (*exoU+/exoS−/exoT+/exoY−*); V7 (*exoU−/exoS+/exoT+/exoY−*); OV (other virulotype group): comprising V4 (*exoU−/exoS−/exoT+/exoY−*), V5 (*exoU−/exoS−/exoT+/exoY*+), V6 (*exoU−/exoS−/exoT−/exoY−*), V8 (*exoU−/exoS−/exoT−/exoY*+), V9 (*exoU+/exoS+/exoT+/exoY*+), V10 (*exoU−/exoS+/exoT−/exoY*+) and V11 (*exoU+/exoS+/exoT+/exoY−*).

**Table 4 pathogens-13-01030-t004:** Factors associated with the most frequent virulotypes of *P. aeruginosa*, with the V3 virulotype used as a reference.

	OV vs. V3	V1 vs. V3	V2 vs. V3	V7 vs. V3
Death, yes vs. no: gOR	0.963	5.12	4.756	0
95% CI	0.092–10.007	1.264–20.733	1.113–20.328	0
*p*-value	0.975	0.022 *	0.035 *	0.99
Clinical origin, CF vs. BSI: gOR	0	0.16	0	5.048
95% CI	0	0.025–0.999	0	0.185–137.166
*p*-value	0.995	0.049 *	0.99	0.337
Susceptibility profile, PDR vs. MDR: gOR	14.89	7	38.79	0
95% CI	1.079–205.387	1.040–47.099	8.319–180.895	0
*p*-value	0.044 *	0.045 *	<0.001 *	0.997
Susceptibility profile, XDR vs. MDR: gOR	10.506	2.893	2.979	0.473
95% CI	1.212–91.053	0.674–12.418	0.800–11.097	0.017–12.862
*p*-value	0.033 *	0.153	0.104	0.658
Susceptibility profile, S vs. MDR: gOR	0	1.885	0	13.423
95% CI	0	0.134–26.434	0	0.579–310.913
*p*-value	1	0.638	1	0.105

gOR generalized Odds radio, *: significant *p* value < 0.05, CI: confidence interval = 95%, CF: cystic fibrosis, PDR: pandrug resistant, XDR: extensively drug resistant, S: sensitive. V1 (*exoU+/exoS−/exoT+/exoY+*)*;* V2 (*exoU+/exoS−/exoT+/exoY−*); V7 (*exoU−/exoS+/exoT+/exoY−*) OV (other virulotype group): comprising V4 (*exoU−/exoS−/exoT+/exoY−*), V5 (*exoU−/exoS−/exoT+/exoY+*), V6 (*exoU−/exoS−/exoT−/exoY−*), V8 (*exoU−/exoS−/exoT−/exoY+*), V9 (*exoU+/exoS+/exoT+/exoY*+), V10 (*exoU−/exoS+/exoT−/exoY*+) and V11 (*exoU+/exoS+/exoT+/exoY−*).

**Table 5 pathogens-13-01030-t005:** Independence test between the different study variables and death outcome.

Variable		Discharged Patients n = 85 (85%)	Death Patients n = 15 (15%)	Total n = 100 Patients	*p* Value
T3SS virulotype	V1	9 (10.59%)	6 (40%)	15 (15%)	0.003 *
	V2	20 (23.53%)	5 (33.33%)	25 (25%)	0.419
	V3	51 (60%)	6 (40%)	57 (57%)	0.149
	V7	2 (2.35%)	0 (0%)	2 (2%)	0.548
	OV	8 (9.41%)	1 (6.67%)	9 (9%)	0.732
Susceptibility profile	PDR	19 (22.35%)	3 (20%)	22 (22%)	0.839
	XDR	39 (45.88%)	11 (73.33%)	50 (50%)	0.049 *
	MDR	27 (31.76%)	4 (26.67%)	31 (31%)	0.694
	S	4 (4.71%)	2 (13.33%)	6 (6%)	0.195
Clinical origin	CF	12 (14.12%)	4 (26.67%)	16 (16%)	0.222
	BSI	73 (85.88%)	11 (73.33%)	84 (84%)	0.400

*: chi-square test significant, *p* value < 0.05; BSI: bloodstream infection; PDR: pandrug resistant; XDR: extensively drug resistant; MDR: multidrug resistant; S: sensitive. V1 (*exoU+/exoS−/exoT+/exoY+*); V2 (*exoU+/exoS−/exoT+/exoY−*); V3 (*exoU−/exoS+/exoT+/exoY+*); V7 (*exoU−/exoS+/exoT+/exoY−*); OV (other virulotype group): comprising V4 (*exoU−/exoS−/exoT+/exoY−*), V5 (*exoU−/exoS−/exoT+/exoY+*), V6 (*exoU−/exoS−/exoT−/exoY−*), V8 (*exoU−/exoS−/exoT−/exoY+*), V9 (*exoU+/exoS+/exoT+/exoY+*), V10 (*exoU−/exoS+/exoT−/exoY+*) and V11 (*exoU+/exoS+/exoT+/exoY−*).

**Table 6 pathogens-13-01030-t006:** Relative risk for death associated with the most frequent genotypes of *P. aeruginosa.*

Death Clinical Outcome per Variable	RR	95% CI	*p* Value
Clinical origin (Ref. BSI)			
CF	3.643	1.181–11.23	0.024 *
Susceptibility profile (Ref. MDR)			
PDR	0.531	0.144–1.955	0.341
S	2.459	0.772–7.836	0.128
XDR	1.499	0.533–4.216	0.443
Virulotype (Ref. V3)			
OV	1.185	0.110–12.77	0.889
V1	3.690	1.259–10.82	0.0174 *
V2	2.907	0.852–9.917	0.0882
V7	0.00000001	0–0.0000001	<0.001 *

RR: relative risk, *: *p* value < 0.05, CI: confidence interval = 95%, BSI bloodstream infection, PDR: pandrug resistant, XDR: extensively drug resistant, S: sensitive. V1 (*exoU+/exoS−/exoT+/exoY+)*; V2 (*exoU+/exoS−/exoT+/exoY−*); V7 (*exoU−/exoS+/exoT+/exoY−*); OV (other virulotype group): comprising V4 (*exoU−/exoS−/exoT+/exoY−*), V5 (*exoU−/exoS−/exoT+/exoY+*), V6 (*exoU−/exoS−/exoT−/exoY−*), V8 (*exoU−/exoS−/exoT−/exoY+)*, V9 (*exoU+/exoS+/exoT+/exoY*+), V10 (*exoU−/exoS+/exoT−/exoY*+) and V11 (*exoU+/exoS+/exoT+/exoY−*).

## Data Availability

All data generated or analyzed during this study are included in this published article and its Appendix A.

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
