# Peer review of "The exoS, exoT, exoU and exoY Virulotypes of the Type 3 Secretion System in Multidrug Resistant Pseudomonas aeruginosa as a Death Risk Factor in Pediatric Patients"

_pathogens, 2024, doi:10.3390/pathogens13121030_

Round 1

Reviewer 1 Report

Comments and Suggestions for Authors

The work presented by Nolasco-Romero et al. characterizes the presence of TSS3 effectors, at gene level, in a series of Pseudomonas aeruginosa isolated from pediatric patients in Hospital Infantil de México Federico Gómez (HIMFG). This issue is of great importance since the presence of these effectors can be related to the severity of the infection and the prognosis of patients. Authors have performed several statistics analysis in order find different associations between these virulotypes and, resistance, clinical origin or mortality.

I find this paper very interesting; however, there are some issues of my concern:

·         Line 32: Pseudomonas aeruginosa must be written in italics. Authors shloud check and be correct this mistake several times in the manuscript (line 36, line 257, etc)

·         All gene names should be written in italics: line 36: exo; line 41 exoU, exoS, exoT and exoY), etc. Authors shloud check and be correct this mistake several times in the manuscript.

·         Line 40: replace blood-stream by bloodstream.

·         Line 46: authors indicate: “Fifteen patients died (CF=4; B=12)” however 4 plus 12 are 16 patients. Please, correct it.

·         Line 48: “the V1, V2 and CF virulotypes are related”, what authors mean with CF virulotype? Is this correct?  CF is not a virylotype, but an origin.

·         Material and methods, collection of biological samples:

o   Line 124: why the strains were preserved at -70ºC instead of -80ºC?

·         Material and methods, Molecular detection of exo-virulotypes through mPCR.

o   In my opinion, the design of this multiplex PCR for the detection of all exo genes in the same reaction is an important fact, and should be highlight in results and discussion.

o   “The PCR products were finally loaded onto a 1% agarose gel” – PCR products are quite small and the size difference is low, maybe a 1.5-2% gel should be more adequate.

·         Material and methods, Multinomial Regression Model Test.

o   Statistical analysis is one of the most important points of this paper, since through this analysis the results and conclusions are obtained. However, it is not clear how this has been carried out. This point is quite confusing. The authors should clarify this part, clearly explaining what data and strains has been used in each analysis, and why. Also, the type of analysis that has been used for each comparison should been sorted out.

o   In this point, and along the paper, “clinical outcome of death” or “death outcome”, should be replaced by “mortaliy”, “mortality rate” or similar.

·         Results, Identification of T3SS virulotypes:

o   Line 210: eleven “presence+/absence- type patterns” should be replaced by “virulotypes”.

o   If the different virulotypes are described in the table 2, it is not necessary to indicate in the text the genes present or absent in each virulotype. For example, line 212: V3 (exoU-/exoS+/exoT+/exoY+), V3 is enough.

o   Table 2: join in the same column n and % as n (%).

·         Results, Associations between multidrug resistance and virulotypes:

o   Figure 1:

§  Figure legend should be placed below the figure.

§  Names or genes should be in italics.

§  This figure is not clear. How was the dendogram in the left constructed (no PFGE or NGS performed…)? Why ther are only4 virulotypes represented? The presence of black and green points is not understood. This figure should be improved or removed.

§  The difference between the two first

·         Tables 3, 4 and 5:

o   These tables should be should formatted to be clearer and shorter.

o   It is not necessary to include the definition of virulotypes in each table.

o   Table 3: what is the significance of the use of V3 virulotype as reference? Authors should detect the statistical significance between different variables without the use of the reference (Fisher or Kruskal-Wallis test??)

·         Discussion:

o   In order to make a correct discussion, authors should compare exoU and exoS prevalence and compare these data with those in references. Indicate that exoT is the most common is not relevant.

o   Authors discuss about the sequence type and the presence of exoU. However, there is not mention to MLST neither in material and methods nor in results. Please, include MLST data in the manuscript or remove this part in the discussion.

o   Authors should deep about the prevalence of exoU in CF or not CF samples, as well as in BSI or not BSI samples.

Author Response

The authors are very grateful to the reviewers for their suggestions and revisions to the work. Undoubtedly, they were of great help in improving the quality of the work.
Reviewer 1
The work presented by Nolasco-Romero et al. characterizes the presence of TSS3 effectors, at gene level, in a series of Pseudomonas aeruginosa isolated from pediatric patients in Hospital Infantil de México Federico Gómez (HIMFG). This issue is of great importance since the presence of these effectors can be related to the severity of the infection and the prognosis of patients. Authors have performed several statistics analysis in order find different associations between these virulotypes and, resistance, clinical origin or mortality.
I find this paper very interesting; however, there are some issues of my concern:
·         Line 32: Pseudomonas aeruginosa must be written in italics. Authors shloud check and be correct this mistake several times in the manuscript (line 36, line 257, etc)
Thanks for your comment, it's done.
·         All gene names should be written in italics: line 36: exo; line 41 exoU, exoS, exoT and exoY), etc. Authors shloud check and be correct this mistakes several times in the manuscript.
Thanks for your comment, the authors, carefully review the text and verify that it will deal with protein (uppercase) and gene (lowercase and italics). Also, It was also necessary to change the italics and lowercase in the title
·         Line 40: replace blood-stream by bloodstream.
Thanks for your comment, it's done.(Line 46)
·         Line 46: authors indicate: “Fifteen patients died (CF=4; B=12)” however 4 plus 12 are 16 patients. Please, correct it.
Thanks for your comment, it's done. (Line 54)
·         Line 48: “the V1, V2 and CF virulotypes are related”, what authors mean with CF virulotype? Is this correct?  CF is not a virylotype, but an origin.
Thanks for your comment, it's done. (Line 56-57)
·         Material and methods, collection of biological samples:
o   Line 124: why the strains were preserved at -70ºC instead of -80ºC?
Thanks. Although the standard temperature for ultra-freezers is -80°C, in the case of our laboratory we seek to reduce energy consumption, heat production and increase the life of the freezers; therefore, the temperature used for cryopreservation of samples is around -70°C, considering that this temperature is sufficient to minimize metabolic activity and avoid sample degradation by carrying out cryopreservation properly (Landor et al; 2024).
Landor, L.A.I., Stevenson, T., Mayers, K.M.J. et al. DNA, RNA, and prokaryote community sample stability at different ultra-low temperature storage conditions. Environmental Sustainability 7, 77–83 (2024). https://doi.org/10.1007/s42398-023-00297-2

·         Material and methods, Molecular detection of exo-virulotypes through mPCR.
o   In my opinion, the design of this multiplex PCR for the detection of all exo genes in the same reaction is an important fact, and should be highlight in results and discussion.
Thanks for the suggestion, we will address it on the line 437-439
o   “The PCR products were finally loaded onto a 1% agarose gel” – PCR products are quite small and the size difference is low, maybe a 1.5-2% gel should be more adequate.
Thank you, on this occasion we considered a 1% agarose gel to make less use of it, since we observed sufficient clarity with this concentration during the standardization of the reaction.
o   Statistical analysis is one of the most important points of this paper, since through this analysis the results and conclusions are obtained. However, it is not clear how this has been carried out. This point is quite confusing. The authors should clarify this part, clearly explaining what data and strains has been used in each analysis, and why. Also, the type of analysis that has been used for each comparison should been sorted out.
At this point, the text has been changed for better understanding (lines 215-248). We appreciate your comments.
o   In this point, and along the paper, “clinical outcome of death” or “death outcome”, should be replaced by “mortaliy”, “mortality rate” or similar.
Thank you very much. In this case, we chose not to use that term because we are not measuring the mortality rate as such, which is defined as the number of deaths from a disease divided by the total population.
·         Results, Identification of T3SS virulotypes:
o   Line 210: eleven “presence+/absence- type patterns” should be replaced by “virulotypes”.
Thank you, it’s done (Line 285)
o   If the different virulotypes are described in the table 2, it is not necessary to indicate in the text the genes present or absent in each virulotype. For example, line 212: V3 (exoU-/exoS+/exoT+/exoY+), V3 is enough.
Thank you, it’s done (Line 287, 288, 289 and 290 )
o   Table 2: join in the same column n and % as n (%).
In line 296, table 2 was changed and the frequency of n and % was merged into a single column. We appreciate your comment.
·         Results, Associations between multidrug resistance and virulotypes:
o   Figure 1:
§  Figure legend should be placed below the figure.
§  Names or genes should be in italics.
§  This figure is not clear. How was the dendogram in the left constructed (no PFGE or NGS performed…)? Why ther are only4 virulotypes represented? The presence of black and green points is not understood. This figure should be improved or removed.
Thank you, it’s done (Line 372)
§  The difference between the two first
·         Tables 3, 4 and 5:
o   These tables should be should formatted to be clearer and shorter.
o   It is not necessary to include the definition of virulotypes in each table.
Thank you very much for the suggestion, but we think that the information on the virulotypes should be present in each table to make the reading easier, even if the description of the virulotypes is in Table 2. In any case, we await the editor's response and if necessary, we have no problem following the observation
o   Table 3: what is the significance of the use of V3 virulotype as reference? Authors should detect the statistical significance between different variables without the use of the reference (Fisher or Kruskal-Wallis test??)
Thank you. The multinomial logistic regression model is appropriate in this context because it allows for detailed multivariate analysis of the relationship between multiple categories of the response variable (virulotype) and various clinical variables, whereas the Fisher and Kruskal-Wallis tests do not have the same ability to handle models with multiple predictors and response variables with more than two levels. Table 3 has been restructured for better understanding (line 356-358). We greatly appreciate your observation.
·         Discussion:
o   In order to make a correct discussion, authors should compare exoU and exoS prevalence and compare these data with those in references. Indicate that exoT is the most common is not relevant.
Thanks for the suggestion, the change is on the line 468
o   Authors discuss about the sequence type and the presence of exoU. However, there is not mention to MLST neither in material and methods nor in results. Please, include MLST data in the manuscript or remove this part in the discussion.
Thanks for the suggestion, we will address it on the line 564-569
o   Authors should deep about the prevalence of exoU in CF or not CF samples, as well as in BSI or not BSI samples.
Thanks for the suggestion, we will address it on the line 634-656

Reviewer 2 Report

Comments and Suggestions for Authors

See attached file.

Author Response

The authors are very grateful to the reviewers for their suggestions and revisions to the work. Undoubtedly, they were of great help in improving the quality of the work. Reviewer 1 This study addresses a crucial area in pediatric infectious disease research, examining the relationship between Pseudomonas aeruginosaT3SS virulotypes and clinical outcomes in multidrug-resistant strains. The focus on the virulotypes ExoS, ExoT, ExoU, and ExoY in the context of cystic fibrosis and bloodstream infections is highly relevant, particularly given the significant mortality risk associated with specific virulotypes. The manuscript presented by a relevant research group in the area is well- structured with clear explanations in each section; however, some areas could be enhanced to increase clarity, impact, and reproducibility.     Introduction • A brief comparison of the infection severity and prognosis associated withT3SS-positive P. aeruginosa infections versus other major bacterial pathogens in pediatric settings could help emphasize the unique importance of these virulence factors in P. aeruginosa. Thanks for the comment, the additional info was added to the line 66-83   • It might be helpful to brieflyaddress the prevalence of these virulotypes in other patient populations (e.g., adults) and outline why pediatric patients are particularly vulnerable to T3SS-mediated virulence. Thanks for the comment, the additional info was added to the line 83-87   • Providing recent statistics on the prevalence of multidrug-resistant (MDR) and T3SS-positive P. aeruginosa strains, particularly in pediatric or high-risk populations, would establish the relevance of the study. Thanks for the comment, the additional info was added to the line  88-103   • Highlighting epidemiological data specific to pediatric populations, such as infection rates or outcomes for P. aeruginosa infections in children with cystic fibrosis (CF) or bloodstream infections (BSI), would contextualize the study. Thanks for the comment, the additional info was added to the line    Material and Methods • Present inclusion and exclusion criteria adopted for sample selection; At this point, the text has been changed for better understanding (lines 215-248). We appreciate your comments.   • According to CLSI news from June 2023, relevant changes to the aminoglycoside (gentamicin, tobramycin, and amikacin) breakpoints were published in CLSI M100-Ed33. Specifically, elimination of gentamicin as a suggested treatment option for P. aeruginosa and exclusion of breakpoints. Therefore, review and provide which breakpoints were considered; Thanks for the comment, the additional info was added to the line 206-209   • Use the correct reference format: Magiorakos et al. [X], not ‘Magiorakos et al.(2012) Thanks for the comment, the change was added to the line 204, 214   • The study should specify that it received approval from an institutional ethics committee, with the committee’s name, approval number, and date. Thereshould be a mention of how informed consent was obtained from parents orlegal guardians, as pediatric patients cannot legally provide consent. It would be beneficial to specify how patient confidentiality was maintained throughout the study, such as by anonymizing patient data or ensuring secure storage of identifiable information. Thanks for the comment, the additional info was added to the line 152-157   Results • Exo genes should be in italics as it may be confused with the protein; Thank you, it’s done    • Kaplan-Meier survival curve or similar visualization could be useful to illustrate the survival trends by virulotype Thank you for your comment. In our case, we do not have data on the patient's evolution over time that would be appropriate to create the curve, so we decided to create the multinomial regression model evaluating all the variables we have.   • Specifying if certain antibiotics are more associated with specific resistance profiles and virulotypes would add depth. Thank you for your suggestion. The relationship between resistance to a specific antibiotic was not evaluated on this occasion, considering the variability of the data we grouped according to the Magiorakos classification; however, in the discussion we addressed the specific relationship of some antibiotics with the T3SS virulotypes.   • Data from antimicrobial resistance is not directly presented and as just used for comparisons. Thank you for your comment. We have added the reference to the appropriate supplementary material on lines 359-360 where the MICs data is located. In addition, we have added information on this topic in Table 3 (line 383).   Discussion • To enhance this comparison, the authors could include a more detailed discussion on regional or age-related differences in T3SS prevalence or outcomes. For instance, discussing differences between pediatric and adult patient populations in terms of T3SS-associated mortality could further underscore the importance of studying these virulotypes in children. Thanks for the suggestion, we will address it on the line    • Include data of other studies discussing the prevalence of uncommon exoU+/exoS+ virulotype; Thanks for the suggestion, we will address it on the line 606-616   • In this work, with the second set of analyses, we also evaluated the presence of the exoT and exoY genes for the purpose of determining patterns between the virulotype in the T3SS and specificclinical variables, e.g., epidemic high-risk clone isolates.’ I found odd that the authors engage in discussion of epidemic high risk clones, but this data is not presented clearly in the discussion. Thanks for the suggestion, we will address it on the line 564-569   • In addition to these limitations, noting potential sampling biases (e.g., the focus on a single hospital or specific patient groups like CF and BSI) could add transparency. Suggestions for future research could also include investigating virulotype prevalence across other demographic groups, examining other virulence factors alongside T3SS, and evaluating potential virulotype-specific therapies or inhibitors. Thank you very much for the suggestion, we will address it on the line 535-543 • A concluding paragraph summarizing the potential clinical applications of the study’s findings would provide a concise takeaway for readers. Emphasizingthat T3SS virulotyping could become a valuable tool in the prognostication and management of pediatricP. aeruginosainfections would underscore the study’s clinical relevance. Thank you very much for the suggestion, we will address it on the line 653-656

Round 2

Reviewer 1 Report

Comments and Suggestions for Authors

Most of the suggestions and revisions have been adressed.

Reviewer 2 Report

Comments and Suggestions for Authors

Thank you for carefully addressing all the feedback provided in the previous round. The revisions made have greatly improved the manuscript, clarifying key points and strengthening the presentation of your findings. I have no additional comments, and I believe the manuscript is now ready for publication.